# Basic Health Service Delivery to Vulnerable Populations in Post-Conflict Eastern Congo: Asset Mapping

**DOI:** 10.3390/healthcare11202778

**Published:** 2023-10-20

**Authors:** Dieudonne Bwirire, Rik Crutzen, Rianne Letschert, Edmond Ntabe Namegabe, Nanne de Vries

**Affiliations:** 1Department of Health Promotion, CAPHRI Care and Public Health Research Institute, Faculty of Health, Medicine and Life Sciences, Maastricht University, 6229 HA Maastricht, The Netherlands; rik.crutzen@maastrichtuniversity.nl (R.C.); nanne.devries@maastrichtuniversity.nl (N.d.V.); 2Maastricht University, 6200 MD Maastricht, The Netherlands; r.letschert@maastrichtuniversity.nl; 3Faculté de Santé et Développement Communautaires, Université Libre des Pays des Grands Lacs (ULPGL), Goma 368, Democratic Republic of the Congo; ntabenamegabe2006@gmail.com

**Keywords:** community participatory asset mapping, asset-based approaches to health, health inequalities, basic healthcare services, community-based healthcare, Democratic Republic of Congo

## Abstract

Populations in post-conflict settings often have increased healthcare needs, residing in settings where basic services needed to maintain good health may be non-existent or hard to access. Therefore, there is a need for better identification and reallocation of resources as part of the post-conflict health rehabilitation effort. Assets can be described as the collective resources that individuals and communities have at their disposal, which protect against adverse health outcomes and promote health status. This study applies an asset-based approach to explore the most optimal design of health services and to identify the resource constraints for basic health service delivery to the most vulnerable communities in eastern Congo. We implemented the asset mapping in two phases. Firstly, we combined a qualitative survey with community walks to identify the assets already present in the communities. Secondly, we conducted group discussions to map out assets that are the core of asset-based community development (ABCD) practice. We finally documented all assets in a Community Asset Spreadsheet. Overall, 210 assets were identified as available and potentially valuable resources for the communities in eastern Congo. Among them, 57 were related to local associations, 23 to land and physical environments, 43 to local institutions, 46 to individuals, 35 to economy and exchange, and only 6 to culture, history, and stories. Drawing upon the findings of the qualitative survey, community walks, and group discussions, we concluded that an important number of resources are in place for basic health service delivery. By activating existing and potential resources, the most vulnerable populations in eastern Congo might have the required resources for basic health service delivery. Our findings support the use of an asset-mapping research method as appropriate for identifying existing and potential resources for basic health services in a post-conflict setting.

## 1. Introduction

War and armed conflict have profound adverse effects on population health [1,2], including disruption to the delivery of basic healthcare services (BHS). Additionally, they deteriorate social determinants of health and exacerbate health disparities [3,4]. Understanding the impact of war and armed conflict on health inequality is essential for developing effective health policies and practices [5]. Without a better understanding of these factors and what can be done about them [6], interventions cannot be effectively targeted and might lack relevance, resulting in misdirected use of resources [7]. To eliminate barriers to basic healthcare service utilization and improve health outcomes, collaborative efforts with the community are crucial. Accordingly, the WHO emphasizes the importance of resources and assets to good health: resilience, capabilities, and strengths of individuals and communities need to be built on and the hazards and risks to which they are subjected need to be addressed [8]. For instance, one important concept at the heart of community approaches is community assets. According to McLean, community assets are “…*the collective resources which individuals, families, and communities have at their disposal that protect against negative health outcomes and promote health and well-being and improve life chances*” [9]. 

There is some evidence to suggest that disadvantaged communities that are more cohesive are also more likely to maintain their health [10,11]. However, it should also be recognized how particularly challenging it can be to mobilize assets amongst the most vulnerable groups, who typically have the fewest personal assets and assets of the communities they live in [12]. Vulnerability is defined as being at increased risk of harm or having reduced capacity or power to protect one’s interests, and vulnerable groups are considered as such because of disparities in physical, economic, social, and health status compared with the dominant population, which makes them more prone to situations of neediness, dependence, victimhood, or helplessness and more in need of services to protect them or enable them to protect themselves [13]. Examples of groups that are particularly vulnerable and disproportionately affected by the impacts of conflict include the civilian population, women, children, internally displaced persons (IDPs), the elderly, and the mentally ill [14].

Another important concept of community approaches is health assets (HAs). Morgan and Ziglio defined HAs as “any factor which enhances the ability of individuals, groups, communities, populations, social systems and/or institutions to maintain and sustain health and well-being and to help to reduce health inequities” [6]. They advocate using Kretzmann and McKnight’s method of rebuilding troubled communities as a practical approach to public health, as it goes far beyond intrapersonal assets to include practically anything that a community identifies as its own that can potentially benefit coexistence, development, and health [15]. 

Healthy community initiatives are better served by “asset-based” approaches (ABAs) than by “need-based” approaches (NBAs). Whereas ABAs invite more creativity in assessment and planning [16], NBAs focus on a community’s needs, deficiencies, and problems. They can have negative effects even when a positive change is intended because they force community leaders to highlight their communities’ worst side to attract resources [15]. In both approaches, hard realities must be faced. Consequently, the incorporation of an assessment of community assets—in addition to health needs—has become a standard best practice promoted by the National Association of County and City Health Officials (NACCHO) and the Public Health Accreditation Board (PHAB) in the United States [17]. ABAs focus on people’s and communities’ capacities, resources, and networks, as well as on their needs [18]. Hence, they are not only “people-centered” but also “citizen-driven” approaches. Furthermore, they are solution-oriented and draw on evidence-based practical frameworks. Although it recognizes the importance of NBAs and ABAs, this paper will mainly focus on ABAs because they involve and transform community members from individual recipients to a collective of empowered citizens and advocates for their community. 

Post-conflict health rehabilitation (PCHR) is an effort to maximize positive outcomes with constrained resources. It implies a prioritization process for the allocation of resources. Because there is no single universally accepted approach to resource allocation, this should be based on the local reality [19]. Although the success of PCHR programs depends on factors that are outside of the immediate scope of a health sector rehabilitation program, its apparent success also depends on local commitment, local capacity, and local assets. Planning for the medium- and long-term development of the health system is essential for successful health system rehabilitation. Therefore, a first step towards addressing this gap is to produce evidence of local community health assets, how they work, and whom they work for. This paper addresses this gap by using an ABA that views population health through a different lens. The approach is about promoting and strengthening the factors that support good health and well-being, protecting against poor health, and building and fostering communities and networks that sustain well-being [20].

### Theoretical Framework

This study employed the asset-mapping approach to identify resources and to mobilize community members for local development purposes [15]. The process provides a good understanding of what is or is not available in dealing with an issue of interest and sets up informed evidence-based planning. Asset mapping, therefore, represents a distinctive process, underpinned by a fundamentally different logic than that of needs assessment. However, in comparison with the rich literature on needs assessment, there is little empirical research to demonstrate how asset mapping leads to health improvement [21]. 

In post-conflict situations, assets are critical to the development of a sustainable and successful health system, as existing resources could be used to rebuild better. Essentially, restoring damaged or destroyed resources and assets also becomes a priority. Unfortunately, for the Democratic Republic of Congo (DRC), despite its actual post-conflict status, the current assets of the population have not been formally identified, and the needs of the population are unknown and have, for the most part, not been effectively addressed [22,23,24]. Therefore, this study aims to explore the most optimal design of health services and to identify the resource constraints for BHS delivery to the most vulnerable populations in eastern Congo through an ABA.

## 2. Material and Methods

### 2.1. Study Settings and Participants

This study was carried out in eastern Congo, where many families have unmet needs, including access to essential resources such as clean water and food. The situation in eastern Congo is particularly relevant because of the continuous combination of armed confrontation, internal displacement, and predatory state structures. The DRC has been deeply fractured by a conflict that can be divided into three episodes. The First War began in November 1996 and ended with the toppling of President Mobutu Sese Seko in May 1997 by Laurent Desiré Kabila. The Second War began in August 1998 and was characterized by the participation of many actors (Angola, Chad, the DRC, Namibia, Sudan, Zimbabwe, and the Maï-Maï- and Hutu-aligned forces) in complex alignments. Four peace agreements—Lusaka (1999), Sun City (2002), Pretoria (2002), and Luanda (2002)—were needed to finally end the war. However, this final agreement has not succeeded in ridding the DRC of violence, especially in the eastern regions, in what can be considered the third episode of the conflict [25]. In this study, the term “eastern Congo” refers to three locations where data were collected from January to March 2021, including locations in South Kivu Province (Bukavu, Katana-Kahungu, Kalehe-Ihimbi, and Walungu), North Kivu Province (Goma, Mushake-Masisi, Bujovu-Nyiragongo, and Kirotshe-Shasha), and Ituri Province (Bunia-Kigonze, Lita-Bahwere, and Rwampara–Shari).

Participants were recruited from the communities of eastern Congo if they were aged 18 years or older, were living in eastern Congo, and had experienced at least one episode of conflict in the last 10 years. The final sample size was determined by data saturation. Participants had to agree to take part in community walking and community-engaged mapping discussions by providing written or verbal consent.

### 2.2. Study Procedures

We conducted asset mapping based on the experiences and perspectives of community members using different qualitative methods with community members. Particularly, we adopted the asset-based community development (ABCD) mapping framework [26], which provides a step-by-step guide on how to identify the assets already present in communities to mobilize them in support of community development. We implemented the mapping in two main phases; below, we describe the details of the various methods used in each phase separately.

The first phase comprised the identification of the assets already present in the community and the proper implementation of asset-mapping processes by combining the asset-mapping survey and community walks in the study at hand:

Asset-Mapping Survey (AMS): A semi-structured interview guide (see Appendix B) was used that was developed with a small number of open-ended questions covering different topics that mainly focused on gaining a better understanding of their assets. These qualitative interviews were considered an appropriate method to explore perspectives on asset-mapping processes and outcomes [27]. As a result, they helped to identify informal leader(s) who should be involved in the community engagement mapping (CEM) discussion, which occurs at the end of a community walk. The semi-structured interview guide was first translated from English into Kiswahili and French and then back-translated into English. Eleven interviews were carried out by two field researchers.

Community Walks: Participants walked through their communities to identify assets and resources as they saw them through their eyes. Each participant was equipped with a blank sheet of paper on which they created a small map that they used throughout the walk to document the assets they identified and key observations. The participants included two field researchers from the Research Initiatives for Social Development (RISD)—a leading organization in training data collection teams, research implementation, and development projects, and community members randomly selected from the three study settings by the RISD researchers during the walk. They were randomly selected in each setting using a list of all avenues from the individually created small maps. A systematic approach was then used to generate a number that was applied as a constant interval to select the next avenue. Additionally, a snowball-sampling strategy was used to ensure that selected community participants would act as recruitment or referral agents for further participation [28]. One hundred fifty-three participants took part in the community walks across the three study settings in eastern Congo (see Appendix C). 

The second phase consisted of the CEM, which is a group asset-mapping event designed to gather feedback from community members about where they live, work, or attend school in the area. The goal is to map community assets that are the core of ABCD practice [29], such as (1) individuals, (2) local associations, (3) local institutions, (4) land and physical environment, (5) economy and exchange, and (6) culture, history, and stories [30,31]. ABCD assumes that assets fit into three major categories that are the asset base of every community, such as individual, associational, and institutional [15]. This also provides a framework for recognizing, mapping, and mobilizing community assets. The one hundred fifty-three participants who also completed the community walks were split into breakout groups of approximately 8 to 12 formal and informal leader(s) per group, which can be described as a focus group around a map. They were then provided with a physical map printout (prepared beforehand by the field researchers on a flip chart) that they used to center their discussion around the location of the assets and discuss the resources in their communities that they have accessed. By doing so, they were able to identify strengths and assets within the community, resources that they accessed outside of the community, and areas that can be improved to become assets. 

During both study phases, communication between community members and field researchers happened smoothly, as the field researchers spoke the local language and were mostly able to familiarize themselves with the community and build mutual trust before explaining the study activities. Additionally, they all had good community networks, had assisted with previous research projects conducted in the same settings, and had spent time in the community familiarizing and liaising with local institutions.

### 2.3. Data Sources

To obtain information on the resource constraints, we built on the existing six classic categories of assets [29] and visualized the available resources for BHS delivery in eastern Congo. Participants’ perspectives about existing assets were collected during interviews and CEM discussions that were conducted in the local languages (Kiswahili or French) by two field researchers who were trained by the first author on the use of the interview and CEM discussion guides. 

### 2.4. Data Analysis

Information gathered from each study setting was transcribed for content analysis [32]. Content analysis [33,34,35] was employed to review the qualitative data stemming from the semi-structured interview transcripts and CEM group discussions to pull out themes that would serve as the unit of analysis [36]. Based on an early discussion with the field researchers, we adopted a sequential approach to data analysis that included both deductive and inductive approaches [37,38]. Deductive coding (pre-set of codes) was used as an organizing framework for the coding process, assuming that the core concepts were in the data based on knowledge of the extant literature on ABCD [39,40], but themes were not anticipated during the design. The main field researcher (B.M.C.) and the first author (D.B.) individually analyzed the transcripts according to this coding process. Inductive coding (codes derived from the data) was used for open coding (notes and headings were written in the text while reading it), creating categories and abstractions [32]. It also means that themes were mainly inductively generated [41], and data were categorized as anticipated from the design of the interview and CEM guides. The coding was conducted manually because data gathering was carried out on a relatively small scale and thus was an appropriate and manageable project to analyze in this manner [42,43,44]. The names and identifiers were removed from the transcriptions. The transcribed notes from each study setting were interpreted in terms of the six categories of core assets being investigated (see Table 1. Sample codes, definitions, and examples). 

To check for accuracy, two field researchers reviewed the main results that were generated from CEM discussions with participants, asking them to add to or comment on the results presented by asking whether they had captured their full reflections. We will further examine the six categories of core assets and discuss the themes generated during data analysis. More importantly, we will also examine the interconnections among assets and find ways to access them. Examples and quotes that illustrate community assets will be highlighted. 

### 2.5. Ethics Statement

Ethical approvals for the study were obtained from the ethics committees from the Université Libre des Pays du Grand Lac in DR Congo (reference number: 003/CE/ULPG/MK/2020) and from Maastricht University in the Netherlands (approval number: FHML-REC/2020/077). In addition to these formal ethical approvals, agreements with local competent health authorities were developed and signed by both the first author and the representatives of the local health authority. All participants were given an information sheet detailing the project and their rights to withdraw from it at any point throughout the duration of the study; each participant signed a consent form to participate.

## 3. Results

### 3.1. Participants’ Characteristics

A total of 164 community members participated in this study; 104 were males (64%) and 60 were females (36%). Their ages ranged from 18 to 69 years old. Additional characteristics of the included participants are presented in Table 2.

### 3.2. Study Phase One: Asset-Mapping Survey and Community Walk(s)

Table 3 displays sample quotes from community members during the asset-mapping survey. Overall, the surveys revealed capabilities that people would like to share with their respective communities, including childcare, elderly care, conflict management, mediation, praying for sick people, agriculture, and gardening. They also revealed that, in general, people care the most about issues related to childcare, family care, security, and food security. 

### 3.3. Phase Two: Community Engagement Mapping 

The Appendix A outlines the number and categories of assets captured and shows how those assets were visualized through the asset map resources sheet (see Appendix A). Approximately 210 assets were identified as available and potentially valuable resources for the communities in eastern Congo. Among them, 57 were related to local associations, 23 to land and physical environments, 43 to local institutions, 46 to individuals, 35 to economy and exchange, and only 6 to culture, history, and stories. These assets are interconnected and may influence each other. Sample quotes from CEM Discussions are presented in Table 4. We further present these findings concerning the aforementioned six classic categories (represented in three settings) of community assets, including the ones outlined below.

Local association assets. These are formal or informal groups of people working together as volunteers to generate collective action. Out of the 210 potentially valuable resources identified in eastern Congo, 57 (27%) were categorized by the community members as local associations. When clustered by type and associations working together for a common purpose, we found that the vast majority of associations are working on the emancipation of women, agriculture, food security, and community development. While exploring questions about how these assets can relate to health inequalities, CEM participants explained the following:
*“…if a member of the local association becomes ill, he/she can borrow money from the association, use it to pay for basic healthcare, and pay it back to the association gradually. As a member of the local association, you can also borrow money and start a small business.”**CEM participant, Walungu*


*“The local associations are not negatively linked to health inequalities, on the contrary, the members of a solidarity association must ensure that one of them benefits from good quality care by making contributions, each according to his or her means, if the sick person is not able to take care of him or herself.”*

*CEM participant, Bukavu*


Land and physical environment assets. These included both the built environment, such as buildings and streets, as well as accessible environmental assets, such as physical spaces, which are environments where information is encountered, gathered, interacted with, and used. Community members use land and physical assets for sports as well as for income-generating activities. Out of the 210 potentially valuable resources identified in eastern Congo, 23 (11%) were categorized by the community members as land and physical environment assets. 

Community members were also able to map land and physical assets to churches, football pitches, guest houses, and small and big markets. Most CEM participants noted additional advantages of physical assets:
*“…young people need fields to practice their sports and discover their talents. But, sports fields are not always ready to be used because rivers flood in many areas. Other physical areas are used as reception centers for victims of conflict where physical and psychological care are offered, and also the same areas are used by the community for income-generating activities.”**CEM participant, Walungu*
*“Sport is very important for everyone’s health, especially young people who need these spaces to train in football and basketball.”**CEM participant, Bukavu*

However, other CEM group participants did not find additional advantages of physical assets, especially their relationship with health inequalities:
*“We don’t think that these places could be linked to health inequalities. We don’t think it’s related.”**CEM participant, Goma*

On the contrary, they were related to something different:
*“Often these physical spaces are only involved in tourism.”**CEM participant, Kigonze*

Local institutions’ assets. These are involved in cataloging both formal and informal institutions. Formal institutions included schools, faith communities, and local government. Informal associations are typically more ad hoc groups, such as sports teams and local associations. 

Institutions’ assets help the community capture valuable resources and establish a sense of civic responsibility. Out of the 210 potentially valuable resources identified in eastern Congo, 43 (20%) were categorized by the community as local institution assets. These could be mapped into primary and secondary schools, high schools and universities (including research centers), and local and national institutions. Participants discussed the need for more local institutions in the community. As CEM participants succinctly put it:
*“Schools must teach children and young people that we are all equal, with or without money, and we all have the right to health care. But also our children are often injured by their peers at school when we cannot afford to take them to the health center.”**CEM participant, Shasha*
*“Local institutions take care of their workers and offer them health insurance. With this, the workers will have better access to health care. Often the people who are best taken care of in hospitals are those who have health care insurance from these institutions.”**CEM participant, Bukavu*

*Individual assets.* Every community member possesses a vast assortment of individual assets, such as skills (e.g., caregiving skills), talents (e.g., teaching) and interests (e.g., sharing of skills), that give the individual a unique perspective to share. A critical part of building a healthier community is finding out which individual assets every community member possesses. Out of the 210 potentially valuable resources identified in eastern Congo, 46 (22%) were categorized by the community as individual assets. These could further be grouped into political assets, healthcare delivery systems assets (traditional and modern), artistic assets, or other skills. CEM participants commented the following:
*“Speaking of health, doctors and nurses are the most concerned, as they are the ones who experience health inequalities in hospitals, health centers, and communities.”**CEM participant, Bukavu*
*“…these people create jobs in our area. If you have a job, you will be able to feed your family and help everyone to be in good health.”**CEM participants, Walungu*

*Economy and exchange assets.* These focus on the assets that can trigger or form the basis of entrepreneurial opportunities as well as existing and emerging business opportunities. Out of the 210 potentially valuable resources identified in eastern Congo, 35 (17%) were local economic assets dealing with money or access. These were further categorized by the community into agriculture, livestock and agro-pastoral, fishing and maritime traffic, trade, and other businesses. CEM participants commented the following:
*“If we don’t have money, we cannot access basic health care. So, we work every day to gain money, buy food and feed our families, buy medicines in case of illnesses, or go to the hospital if very sick.”**CEM participant, Ihimbi*
*“We all know that inequalities in health are much more linked to the economy, as soon as you don’t have the money you can’t be taken care of.”**CEM participant, Shasha*

Culture, history, and stories assets. Each community is unique, with its own unique history, culture, and local flavor. From the stories, people emerge who have shown commitment and leadership in the past or who are currently taking a leadership role. This should be followed by bringing together a group of committed individuals who are interested in exploring the community’s assets, identifying opportunities, and leading developmental action. Out of the 210 potentially valuable resources identified in eastern Congo, 6 (2%) were categorized as culture, history, and stories by the community. These were mapped into cultural centers and meeting grounds for cinemas.

### 3.4. Mobilizing Identified Assets for Collective Action

Participants also identified underdeveloped factors that could be associated with BHS delivery to vulnerable populations in eastern Congo. By considering those other factors when addressing health issues, communities have “*an opportunity to mobilize existing strengths and resources*” [45]. Guided by the community participatory approach, the mobilization process helps community members to better understand which HAs exist in their respective communities and connect about how these assets might be used to better rebuild BHS delivery to the most vulnerable populations.

## 4. Discussion

This study applies an ABA to explore the optimal design of health services and to identify the resource constraints for BHS delivery to the most vulnerable populations in eastern Congo. Following recommendations by the Marmot Review [46], “*Effective local delivery requires effective participatory decision-making at the local level.*” To this end, asset mapping is a first step towards effective local delivery of BHS, as it produces evidence of the local community HAs, how they work, and whom they work for, moving beyond an NBA. 

We describe existing assets in line with the six classic categories of assets that were preidentified from the literature and what they represent for the community. Among them, there were 57 related to associations, 23 to physical environments, 43 to institutions, 46 to individuals, 35 to local economies, and 6 to culture, history, and stories. More importantly, most of the participants indicated that these assets are interconnected, meaning they have a close or shared relationship and in that way influence each other. 

Local association assets. In eastern Congo, local associations dominated the social assets landscape. We found that the vast majority of these associations are working on women’s issues, agriculture and food security, and community development. They create a network of care and mutual support in times of crisis, and that has the potential for a rapid response to local problems. This suggests that local associations promote mutual support in the communities and help people to collectively overcome protractive crises, as well as promoting networks and illustrating how they can facilitate access to resources. It also highlights the power of connections and social capital in the creation of health. These associations are the vehicles through which the community’s assets can be identified and connected in ways that multiply their effectiveness [47]. Elsewhere, similar work has highlighted how local organizations in Mozambique, Bangladesh, India, and Malawi can identify health priorities and work with facilitators to design effective interventions [48]. If basic healthcare is to be effectively delivered in eastern Congo, it will require good collaboration and trust with existing local associations. In this specific context, existing social assets appear to be particularly relevant given their focus on improving BHS delivery to disadvantaged communities.

Land and physical environment assets. Physical assets include the natural resources and infrastructure of a community [49]. Participants also highlighted valuable physical assets, including land, natural resources, and built environment, that can contribute to efforts to improve BHS delivery to the most vulnerable. Particularly, two types of physical assets support BHS delivery—sports and recreational activities—as most participants emphasized the role of physical assets in providing them access to sports facilities and income-generating activities. On the contrary, a limited number of participants could not find a connection between physical assets and health. This finding provides insight into the role physical assets play in supporting basic healthcare to the community of eastern Congo. Given the specific context, these physical assets may create a safe physical environment and increase the social cohesion that supports basic healthcare service delivery. These findings are consistent with other studies suggesting that physical assets are important components of healthy and livable communities [50,51]. However, low-income communities often face health disparities related to a lack of physical space that could enable BHS delivery [51].

Local institutions’ assets. Our findings also indicate that resources were listed under the institutional category, where institutions referred to assets related to public service provisions, such as facilities providing resources related to health, safety, and general welfare. During the asset mapping, the participants generated evidence of existing institutional resources that could provide valuable environments for BHS delivery. One of the institutional assets that support BHS delivery in eastern Congo is the Bureau Diocésain des Œuvres Médicales (BDOM). It aims to improve the health status of the population in South Kivu by offering medical and technical services to the broader community. This is particularly relevant in this context, where more work is required to understand how the basic capacity for health service delivery can be built and equitably distributed. These findings are supported by previous work suggesting that institutional assets can contribute to improving BHS delivery [52,53].

Individual assets. Furthermore, most participants implied that individual assets (also referred to as personal or human capital) are valuable resources to their communities and should be viewed as co-producers of their own and their community’s well-being. Particularly, the process of developing capacity is more important than the presence or absence of specific resources in the community. Specifically, it includes communities’ power to enhance their health. Finally, participants critically discussed the individual’s capacity to properly use these resources and the relationship between individuals and community groups, as this was mainly perceived as an indicator of successful collaboration.

Our findings provide insight into the various roles of human capital in supporting BHS delivery to the most vulnerable populations. One of the key roles is the ability to work across boundaries between traditional and professional healthcare providers as well as between individual and community-focused healthcare delivery. Findings from a previous study [54] further support the affirmation that assets play a key role in providing every individual with a sense of capacity and purpose. In the specific context of eastern Congo, future research should explore the role of individual capacity-building compared to the group capacity-building approach and select what would fit the purpose of BHS delivery.

Culture, history, and stories assets. Cultural mapping is instrumental in promoting self-awareness and understanding of the social diversity within a community. It was undertaken as an initial step of cultural planning—thus, before deciding how to support and promote cultural assets. It triangulates a range of data on resources such as artists, organizations, festivals, and other communities of interest. In this respect, it is a valuable starting point for promoting community health among the most vulnerable populations. In this study, community members spoke about particular cultural assets as a possible way to deliver basic healthcare services. Participants identified a very limited number of cultural assets (including cultural centers and meeting grounds for cinemas), suggesting that there is very little space for cultural mapping activities that might provide a full picture of the cultural landscape. However, cultural assets cannot be captured by one-off consultations and require an approach that may be the first step in a long journey toward cultural sustainability [55]. This is particularly relevant in the context of eastern Congo, as a cultural asset is an essential asset that is often overlooked in the Global South [49].

Economy and exchange assets. Finally, our findings indicate that resources were listed under the economic category of assets, where “economic” refers to businesses that can provide financial support to the most vulnerable populations. During the asset mapping, the participants generated evidence of existing economic resources, showing community members how well local economic resources are maximized for local economic benefit. This way, the most vulnerable people are empowered with economic concepts through this practically oriented discussion. This finding provides insight into the role economic resources play in supporting basic health and quantifies which resources it can leverage to improve BHS delivery to the most vulnerable populations, which can then be turned into targeted and implementable strategic plans. The findings of this study are consistent with other studies on the association between health and wealth [45,56,57], suggesting that high economic growth leads to investments in human capital and health advancement and good population health leads to more labor productivity and economic growth. Although the study participants were able to establish a relationship between BHS delivery and the local economy, the economic theory fails to recognize the concept of community at all, because although the economic theory demands the free mobility of both labor and capital, the concept of the community gets in the way of this free flow [58]. This is a potential limitation of community participation, which is one of the strategies of an ABA.

### Strengths and Limitations

Strengths of the current study include the participation of the local community in the asset-mapping process. This helps to build trust between community members and develop the capacity to recognize already existing assets (tangible) and identify potential assets that may not be apparent (intangible) [15,45]. Unique to the process is the application of asset-based principles for identifying community assets (by the community members themselves rather than by someone from outside the community) that hold the potential to improve the design and delivery of basic healthcare services meant to improve BHS delivery. This suggests that asset mapping identifies and builds on assets and capacities already in a community rather than merely listing problems to be resolved, which emphasizes community strengths as having the greatest potential to advance sustainable development at the community level and effect lasting change. 

A strength of particular interest was the full involvement of the community-engaged participants. Importantly, this participatory approach increases the validity of asset mapping, particularly when the field researchers are not members of the local community being studied [59]. Furthermore, various qualitative methods for data collection from the experiences and perspectives of community members were used, which provide more varied answers, broaden the information obtained, and give it a high level of validity [60].

In terms of limitations, first, given the fragile context of eastern Congo, not all valuable assets may have been identified, and those that have been highlighted may not be useable [61]. Only identifying valuable assets available to the community in eastern Congo would not be sufficient; therefore, future research could focus on a post-asset-mapping phase consisting of different activities (eventually leading to interventions) that could set the foundation for BHS delivery and enhance the quality of life in the community. 

Second, the study included a limited number of community members who were able to participate in both study phases. This is reflective of the reality of mapping the assets available. More importantly, our findings are still based on a large body of qualitative data. Finally, because the study was conducted in three randomly selected settings, questions about transferability to the rest of the DRC may arise. 

Our findings are likely to be context-specific, and their generalizability outside of eastern Congo was not considered under the current asset mapping. However, our findings can inform policies and programs to identify and mobilize resources in a post-conflict setting. This implies that future research should focus on how asset mapping could be used in these contexts to empower the local community to advocate for better quality of healthcare. Our study demonstrates that despite decades of protracted conflicts and their overall impact, there are existing assets that should be used as the foundation for an optimal healthcare delivery system. To be successful in this environment, all stakeholders (including development initiatives) should focus on how asset mapping could be used in these contexts to empower the local community to advocate for better quality of healthcare.

## 5. Conclusions

This paper describes how community members have used asset mapping to identify existing and potential resources and outlines how they could be applied in future research. From this perspective, asset mapping shows that it has the potential to contribute to ongoing discussions about how to improve BHS delivery by enhancing participation, capacity, and value within communities. By activating the existing and potential resources, community members will have the required resources for BHS delivery. These findings support the use of an asset-mapping research method as appropriate for identifying existing and potential resources for BHS delivery in a post-conflict setting. Given the particular context of eastern Congo, starting from the six classic assets provides a unique framework for the categorization of the existing and potential assets and supports the development of a straightforward BHS delivery system for the most vulnerable populations.

## Figures and Tables

**Table 1 healthcare-11-02778-t001:** Sample codes, definitions, and examples.

Codes	Definitions	Examples Identified in Transcripts	Creation
Social assets	Connections or relationships formed between individuals who live in the community and their unique skills and contributions	Friends, neighbors, family members, etc.	Before analysis, subject to modification during analysis
Physical assets	Physical and virtual spaces, including land, natural resources, and built environments, where information is encountered, gathered, interacted with, and used	Pathways for walking	Before analysis, subject to modification during analysis
Institutional assets	Associations of groups of people who come together around a common purpose	Grocery stores, businesses, schools, and other private or government entities	Before analysis, subject to modification during analysis
Individual	A person belonging to a community, group of relatives, or other people who have capabilities, abilities, and gifts	This is about individuals and relationships.	Before analysis, subject to modification during analysis
Local economy and exchange	Representing the monetary conditions of a community and its people. It also provides information about the business economy of a community.	Local people are working, running their businesses, and purchasing from local stores. Local people and businesses invest in the community.	Before analysis, subject to modification during analysis
Culture, history, and stories	An overview of relevant components that should be found in the community	Cinemas, performing arts venues, theaters, live music venues, cultural events, festivals, sporting events, family events, heritage sites, and attractions.	Before analysis, subject to modification during analysis

**Table 2 healthcare-11-02778-t002:** Participant categories by study setting in eastern Congo.

Eastern Congo Provinces	Study Site	Asset-Mapping Survey	Community Walk	Community-Engaged Mapping (CEM)
North Kivu	Location			
	Goma	1	11	11
	Mushake (Masisi)	1	10	10
	Bujovu (Nyiragongo)	1	10	10
	Shasha (Kirotshe)	1	10	10
South Kivu	Location			
	Bukavu	1	15	15
	Ihimbi (Kalehe)	1	12	12
	Kahungu (Katana)	1	12	12
	Walungu	1	15	15
Ituri Province	Location			
	Kigonze (Bunia)	1	30	30
	Lita (Bahwere)	1	13	13
	Rwampara (Shari)	1	15	15
Number of participants		11	153	153

**Table 3 healthcare-11-02778-t003:** Sample quotes from community members during the asset-mapping survey (AMS).

Questions	Quotes
What are the skills or capabilities that you would like to share with your community?	“I love children because God gave me the grace to have twins three times, So I have a great affection for children. I also live with my parents, they are old and I take care of them.” AMS respondent, Bujovu “I pray for the sick who suffer from evil spirits, I am also a teacher and principal of a primary school.” AMS respondent, Ihimbi “I have lived with my grandmother since the death of my parents, and as soon as she gets sick, I take care of her, using traditional medicines.” AMS respondent, Kahungu “I often intervene when there are conflicts in the community, by directing the victim to the health center or the police. I also take care of victims of sexual violence.” AMS respondent, Shasha “I am more interested in gardening and childcare, these are the two jobs I do the most when I am not in the fields so I am at home with my children.” AMS respondent, Walungu “Childcare, I am a mother in the SOS Bukavu youth home, from house 11 in the Pageco district. We also provide orphans with advice like their biological mothers would do.” AMS respondent, Bukavu “I take care of the sick because I have been a nurse at the Mama RITA dispensary for 4 years, and this is the job I love the most.” AMS respondent Goma “Children’s health and education.” AMS participant, Bahwere “I am very interested in agriculture.” AMS respondent, Rwampara “Advise, when there are problems or a need in the community.” AMS respondent, Kigonze
What do you care the most about?	“Issues relating to children care, and teenage pregnancies. Children live with difficulty and often without a stable family and do not grow up in the warmth of a home. This phenomenon is often at the root of street children and banditry.” AMS respondent, Bujovu “Children’s education, these children are our future.” AMS respondent, Ihimbi “Issues relating to the elderly.” AMS participant, Kahungu “Teenage pregnancies. I would like to help them if at least we had a space or give them advice, many of them are uneducated.” AMS respondent, Mushake “Teenage pregnancies, education, and nutrition, because the number of malnourished children is increasing day by day.” AMS respondent, Shasha “Issues relating to children, and family. When the children are well-educated everything goes well in the community over time.” AMS respondent, Walungu “Child and family care.” AMS respondent, Bukavu “Saving the lives of sick people and those in need, inequalities in health that we observe every day. It would be a very big process to eradicate this situation because we are not all financially equal, it is difficult to be all treated the same way.” AMS respondent, Goma “Security in the region but also Breeding.” AMS respondent, Bawerhe “Issues relating to children care and malnutrition.” AMS respondent, Rwampara “issues relating to children, their food, and their health.” AMS respondent, Kigonze
Which associations do you belong to?	“I am a member of the association of resellers. I am also a member of the Church.” AMS respondent, Bujovu “I am a member of different associations such as AVEC, Groups of men, the local development committee, and the Church.” AMS respondent, Ihimbi “I belong to different associations such as AVEC, GADIP, and youth parliament.” AMS respondent, Kahungu “I am part of the Solidarity for Social Promotion and Peace, AVEC, AJVDI, and the CEPAC church.” AMS respondent, Shasha “I am a member of the Church but also a member of AVEC.” AMS respondent, Walungu “I don’t belong to any associations because the work I do takes up a lot of my time.” AMS respondent, Bukavu “For the moment, I am only a member of the church.” AMS respondent, Goma “I am a member of the footballer’s committee and a member of the motorcycle drivers committee (ATAMOI).” AMS respondent, Bahwere “I am a member of the motorcycle drivers committee of Ituri (ATAMOI).” AMS respondent, Kigonze
Who else do you know (and share the same passion with) in the community?	“The local development committee and the Mutual solidarity fund.” AMS respondent, Ihimbi “The people who help us are the traditional healers.” Kahungu “The MMR association (Maternity at lower risk), which helps pregnant women.” AMS respondent, Shasha “In our church, there are people who educate others on issues concerning their health and how to take care of themselves.” AMS respondent, Walungu “There are many people who have the same passions as me, women who take care of orphaned children.” AMS respondent, Bukavu “I know other nurses I work with, and maybe in the future we will no longer be able to observe these inequalities.” AMS respondent, Goma “Motorcycle taxi drivers.” AMS respondent, Kigonze

**Table 4 healthcare-11-02778-t004:** Sample quotes from CEM Discussions.

Community Assets Categorization	Sample Quotes
Local Associations	“Educating members of these associations on how to rebuild a community, and how people should live together, Bujovu” “These associations can help by teaching their members health-related subjects. Kahungu” “These associations are not involved in health issues, therefore they cannot be linked to health inequalities. Shasha” “Most of these associations are involved in agriculture. Others provide funding to their members as they look for this money to be well taken care of in health centers and hospitals in case of illness. Ihimbi” “For example, AVEC makes funds available for the payment of medical care of its members. The people who do not participate in AVEC are the most impacted and do not have access to healthcare, Mushake” “There are advantages of being a member of these associations: …if a member becomes ill, he/she can borrow money from these associations, use it to pay for basic healthcare, and pay it back to the association gradually. As a member of these associations, you can also borrow money and start a small business. Walungu” “The local associations ensure that their members benefit from good quality care by making financial contributions. Bukavu” “…most of these associations are non-profit, they are much more interested in community development, and there are not immediately focussing on issues related to health inequalities. Goma” “These local associations can help us in case of illness or any other problem, Kingoze” “In case of illness, community workers will bring the patient to the hospital where she/he will be treated first, Bahwere”
Land and Physical Assets	“Often young people are educated about sexuality at the youth corner. In our community, the youth club organizes football tournaments, Ihimbi” “Sport is very necessary for everyone’s health, especially young people who need these spaces to train in football and basketball. Bukavu” “Our young people need fields to practice their sports and discover their talents. However, sports fields are not always ready to be used because rivers flood in many areas. Other physical areas are used as reception centers for victims of conflict. So, physical and psychological treatment are offered in these areas but also the same areas are used by the community for income-generating activities. Walungu” “We don’t think that these places could be linked to health inequalities. We don’t think it’s related. Goma” “No links with health inequalities here. Bahwere” “They are not related to these health inequalities. Shasha” “Often these physical spaces are only involved in tourism. Kigonze”
Local Institutions	“We receive subsidies from these organizations, such as agricultural inputs, farming tools for field work, seeds, and sometimes food products. (Flour, cooking oil…)—Ihimbi” “…they also give us farming tools, and seeds for our fields, but also training in agro-pastoral, nutrition, health, and sanitation. Kahungu” “Temperance heals people for nothing, many go to temperance before going to the hospital. Mushake” “Schools teach children and young people that we are all equal, and we all have the right to health care. But also our children are often injured by their peers at school when we cannot afford to take them to the health center. Shasha” “People who work for local institutions are often better placed and better served in our community, compared to those who live only from agriculture. Walungu” “Local institutions take care of their workers and offer them health insurance. With this, the workers will have better access to health care. Often the people who are best taken care of in hospitals are those who have health care insurance from these institutions. Bukavu” “In our country, there is no such thing as health discrimination, even for the Lendu. Bahwere” “…neighborhood offices look after our safety while schools provide education. Unfortunately, local institutions cannot help us in addressing our health needs. Kigonze”
Individuals	“…are willing to help the community but as individuals, they lack the means. If they were supported by the state, they could help the community better, Bujovu” “…young people are our future, and need to be educated so that they can become nurses and doctors in the future, Bujovu” “These individuals give subsidies to families, often in the form of food products (flour, oil), Kahungu.” “Speaking of health, nurses and doctors are the most concerned, as they are the ones who experience these inequalities in hospitals and health centers. Bukavu” “Nurses and doctors are often linked to these health inequalities because they are often the ones who do not take care of the patients properly, and there is often negligence among the nursing staff. Goma” “…they help us in different situations like illness, education of children, advice, etc. Ihimbi” “…these people help us by paying either part of the bill or all of it for the most vulnerable, especially widows and orphans. Shasha” “…these people create jobs in our area. If you have a job, you will be able to feed your family and help everyone to be in good health. Walungu” “…these are just good people in our community, the chiefs protect us well and the MPs assist us. Bahwere” “Sometimes these people assist us in cases of serious illness or bereavement. Kigonze”
Local economy and exchange	“If we don’t have money we cannot access basic health care. So, we work every day to gain money, buy food and feed our families, buy medicines in case of illnesses, or go to the hospital if very sick, Ihimbi” “It’s what sustains us, it doesn’t just help us with accessing basic health care, but it also helps us pay for our children’s education, Kahungu” “With these activities, we have money that allows us to take care of ourselves in certain situations. Mushake” “We all know that inequalities in health are much more linked to the economy, as soon as you don’t have the money you can’t be taken care of. Shasha” “These activities allow us and our families to survive. We have no other help. Here in our area, if you don’t do one of these activities, you will starve. And we all know that good health is linked to good nutrition. Walungu” “Financial means are often at the root of health inequalities. Those with the greatest financial means are well taken care of, and those without are neglected. Bukavu” “These activities help to fight against health inequalities because when you have the money you are treated very well in hospitals, and when you don’t you are neglected and sometimes you are not even welcomed or taken care of without paying the deposit in some hospitals. Goma” “Having a market next door or shops helps us a lot with supplies because of the proximity, Kigonze”
Culture, history, and stories.	(-)

## Data Availability

Data generated during the present study cannot be shared due to issues of participants’ privacy and confidentiality.

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
