# Peer review of "Basic Health Service Delivery to Vulnerable Populations in Post-Conflict Eastern Congo: Asset Mapping"

_healthcare, 2023, doi:10.3390/healthcare11202778_

Round 1

Reviewer 1 Report

This article is highly relevant and provides deep insight into the issue of underserved populations like those in the DRC. The authors are obviously passionate about the topic, and have a robust expertise and context-knowledge that adds rigor to this article. The method is strong in that it is mixed-method- " a qualitative survey with community walks to identify the assets already pre- 18 sent in the communities. Secondly, we conducted group discussions to map out assets that are the 19 core of Asset-Based Community Development (ABCD) practice. We finally documented all assets 20 in a Community Asset Spreadsheet." This is a well-done piece. 

Suggested changes:

Abstract: "Populations with healthcare needs often reside in post-conflict settings where basic services needed to maintain good health may be non-existent or hard to access" ...consider changing to "populations in post-conflict settings often have increased healthcare needs, residing in settings where..." or add the word "dire healthcare needs". The way it is written now, it overlooks that all populations have healthcare needs

2. Define assets early on and or give examples even in the abstract

3. Needs a good edit overall, including checking for consistant writing style. "“…the collective resources which individuals and communities have at their disposal, 48 those that protect against negative health outcomes and promote health and well-being and improve 49 life chances” [9]. 50 There is som" is in italics but other quotes are not. 

4. The 'Asset Mapping' sub-section needs to be cleaned up a bit and maybe merged with Methods if the authors agree with this? Maybe put it under a title of 'Theoretical Framework'

5. Good job with the method section. Very detailed. Line 215- "All participants were given an infor- 214 mation sheet detailing the project and their rights to withdraw from it within the duration 215 of the study; each signed a consent to participate." should be "each participant signed a consent form to participate"; did all of them sign or did some give verbal consent?

6. In line 219, you write "A total of 164 community members participated in this study, 104 were males (64%) 219 and 60 were females (36%)." What was the attrition rate if any? 

7. Line 230- "Table 3. outlines"- take out the period; small punctuation issues throughout

8. Line 302 section 'Individual assets.' needs some reworking. It is a bit disorganized in thought and hard to follow. 

9. Line 340 section 'Mobilizing Identified Assets for Collective Action' is wonderful. Would love to see more on this. 

10. "We describe existing assets in line with the six classic categories of assets that were 356 preidentified from the literature, and what they represent for the community. Among 357 them, there are 60 associations, 24 physical environments, 43 institutions, 46 individuals, 358 32 local economies, and 6 with culture, history, and stories. More importantly, the partic- 359 ipants found that these assets are interconnected," is strongly written. Just remember to avoid confirmative statements like "the participants found" signaling that all the participants found this to be the case; would revise to "most participants indicated/implied" etc.

11. Wonderful finding: ". We found 362 that the vast majority of these associations are working on women's issues, agriculture 363 and food security, and community development. They create a network of care and mu- 364 tual support in times of crisis, and that has the potential for a rapid response to local prob- 365 lems. This suggests that local associations promote mutual support in the communities 366 and help people to collectively overcome protractive crises, also promoting networks, and 367 illustrating how these can facilitate access to resources."

12. Line 443 needs punctuation

13. Good Strengths and limitations section; maybe elaborate more on why this article contributes to the topic and moreover to the case study of eastern Congo and how development initatives can help support this. Why is it valuable to practitioners? How can these findings lead to changes?

English is good; just needs some editing and help with standard formatting. It may be good to use a professional editor. 

Reviewer 2 Report

The issue addressed by the study is relevant. The methodology used is suitable and the manuscript is well structured. However, authors should:

Abstract

-       Check the number of all assets in the study.

Introduction

-       Provide the definition of vulnerable group in the study.

Materials and Methods

Study Settings and participants

-       Provide the study area map, data collection period, sampling methods, and sample size.

-       Provide a little more contextual information about episode of conflict.

Results

Participants' Characteristics

-       Provide the other participants' characteristics including occupation and socio-economic status.

Study Phase One: Asset Mapping Survey and Community Walk(s)

-       Provide the Table 2 in this sub-heading.

Phase Two: Community-Engagement Mapping

-       Check the number of all assets in the study.

Discussion

-       Each paragraph should contain sub-headings, which can enhance readability.

-       An explanation of the basis on which the other issues for understanding the burden of mental disorders in post-conflict, political strategy, and national sustainable development strategy.

Reviewer 3 Report

This is very useful approach to improve access to healthcare in a country with limited resources and especially a war-torn area. Some comments are found below:

line 31 Because there is no universally correct approach to resource allocation, this 81 should be based on local reality CHANGE TO single universally accepted approach

It is unclear from the text how the communication between field researchers and community participants was conducted, please explain. I assume some of the authors of this work speak the local languages?

In terms of ethics, were participants who were not able to read or write excluded from the study? If not, how was their participation facilitated? If yes, please discuss the implications of this exclusion.

Please elaborate on the random selection of the community participants. Were population registers used? Did any invited community residents refuse to participate to the study? Describe this in reference to the potential of generalization of results.

It would be very useful to indicate some practical actions following the positive results of the asset mapping to empower local residents to receive better quality of health care.

Some minor changes are required
